# Multifunctional Hydroxyapatite/Silver Nanoparticles/Cotton Gauze for Antimicrobial and Biomedical Applications

**DOI:** 10.3390/nano11020429

**Published:** 2021-02-08

**Authors:** Mohamed M. Said, Mohamed Rehan, Said M. El-Sheikh, Magdy K. Zahran, Mohamed S. Abdel-Aziz, Mikhael Bechelany, Ahmed Barhoum

**Affiliations:** 1Chemistry Department, Faculty of Science, Helwan University, Helwan, Cairo 11795, Egypt; za3balawyscience@gmail.com (M.M.S.); zahranmk@science.helwan.edu.eg (M.K.Z.); 2Department of Pretreatment and Finishing of Cellulosic Based Textiles, Textile Industries Research Division, National Research Centre, 33 Bohoth Street, Dokki, P.O. Box 12622, Giza 12522, Egypt; rehan_nrc@yahoo.com; 3Nanomaterials and Nanotechnology Department, Advanced Materials Division, Central Metallurgical R&D Institute (CMRDI), P.O. Box 87 Helwan, Cairo 11421, Egypt; saidelsheikh@cmrdi.sci.eg; 4Microbial Chemistry Department, Genetic Engineering and Biotechnology Division, National Research Centre, 33 Bohoth Street, Dokki, P.O. Box 12622, Giza 12522, Egypt; mohabomerna@yahoo.ca; 5Institut Européen des Membranes, IEM UMR 5635, Université de Montpellier, CNRS, ENSCM, 34090 Montpellier, France; mikhael.bechelany@univ-montp2.fr; 6School of Chemical Sciences, Dublin City University, Dublin 9, Ireland

**Keywords:** medical textiles, plant extract synthesis, ginger oil, brilliant colors, UV protection, antibacterial textiles, wound healing

## Abstract

Medical textiles have played an increasingly important protection role in the healthcare industry. This study was aimed at improving the conventional cotton gauze for achieving advanced biomedical specifications (coloration, UV-protection, anti-inflammation, and antimicrobial activities). These features were obtained by modifying the cotton gauze fabrics via in-situ precipitation of hydroxyapatite nanoparticles (HAp NP), followed by in-situ photosynthesis of silver (Ag) NPs with ginger oil as a green reductant with anti-inflammation properties. The HAp-Ag NPs coating provides good UV-protection properties. To further improve the HAp and Ag NPs dispersion and adhesion on the surface, the cotton gauze fabrics were modified by cationization with chitosan, or by partial carboxymethylation (anionic modification). The influence of the cationic and anionic modifications and HAp and Ag NPs deposition on the cotton gauze properties (coloration, UV-protection, antimicrobial activities, and water absorption) was thoroughly assessed. Overall, the results indicate that chemical (anionic and cationic) modification of the cotton gauze enhances HAp and Ag NPs deposition. Chitosan can increase biocompatibility and promotes wound healing properties of cotton gauze. Ag NP deposition onto cotton gauze fabrics brought high antimicrobial activities against Candida albicans, Gram-positive and Gram-negative bacteria, and improved UV protection.

## 1. Introduction

Medical textiles include all textile materials used for first aid, clinical, surgical, and hygienic purposes, and are among the most dynamic technical textile sectors. Medical textiles can be categorized as: (i) non-implantable, e.g., textile materials for wound healing; (ii) implantable, e.g., sutures, artificial skin, replacement of valves, blood vessels; (iii) extracorporeal (e.g., artificial liver) devices; and (iv) healthcare and hygiene products (e.g., masks) [1]. As dynamic and complex biomechanical processes take place in the human body, medical textiles should be non-toxic, non-allergenic, and non-carcinogenic. Medical textiles used in wound healing, band-aids, sutures, and sanitary towels must be made of biocompatible materials with excellent resistance to microorganisms, sufficient hydrophilicity, and high air permeability [2,3]. Many researchers have been trying to develop textile materials that can block microbial growth [4], using three main approaches: (i) addition of antimicrobial compounds in the textile fibers [5]; (ii) grafting of antimicrobial compounds on the fiber surface [6]; and (iii) coating of the fiber surface with antimicrobial metal and metal oxide nanoparticles (NP) [7,8,9]. Different antimicrobial compounds have been investigated, for instance, quaternary ammonium compounds [10], triclosan [11], metal salts [9], polybiguanides [12], and even natural polymers, such as chitosan [13].

By inhibiting bacterial growth, antimicrobial textiles can avoid germ spreading on the medical textiles, thus preventing the pathogens to get in contact with the human body. However, the antimicrobial agents present in such textiles can be leached during washing because most of them are water-soluble [14], such as quaternary ammonium and nonylphenols that are commonly used but can harm aquatic life. Less water-soluble or non-leaching antimicrobial compounds have been also tested, for instance, metal NPs that show good stability and antibacterial properties. Silver (Ag) NPs are particularly interesting because of their wide spectrum antimicrobial activity (bacteria, fungi, and viruses) [15]. Moreover, due to their specific features, Ag NPs could also be used in drug delivery strategies [16], in the electrodes for detection and diagnosis platforms [17], coatings for medical devices, tissue restoration, and therapeutic products with the improved performance [18]. Medical textiles coated with Ag NPs for antimicrobial activity have been previously described [19]. However, the release of Ag NPs and Ag^+^ ions during their use is a crucial issue because it may decrease the textile antibacterial effect over time, and may also represent a danger for the patient’s safety and the environment. Therefore, many studies have tried to understand how Ag NPs exert their antimicrobial activity and their possible toxic effects on humans, particularly because Ag NPs are the most frequent choice of antimicrobial material for medical textiles [20]. Ag NPs antibacterial activity towards Gram-negative (G−) and Gram-positive (G+) bacteria is mediated by the production of reactive oxygen species [8] that lead to damage of the bacterial cell proteins and DNA, and ultimately to the destruction of the bacterial membrane [21].

Hydroxyapatite (Ca_10_(PO_4_)_6_(OH)_2_) (HAp) is a ceramic that is biocompatible with bone and is therefore used for orthopedic and dental implants. Although HAp displays low biodegradability (Ksp ≈ 6.62 × 10^−126^) compared with other calcium phosphate materials, its chemical formula is similar to that of the bone mineral component and is considered one of the few materials that can promote bonding osteogenesis [22]. Synthetic HAp precursors are obtained using different ceramic processing routes, such as wet precipitation [23], sol-gel synthesis [24], hydrothermal synthesis [25], microwave synthesis [26], and ultrasonic synthesis [27]. HAp can be coated with antimicrobial NPs to inhibit postoperative infection. Moreover, bacterial growth can be inhibited by integrating HAp with hydrogel and antibiotics [28]. HAp has been combined with different polymers to improve its biological properties, such as chitosan, gelatin, agarose, collagen, cellulose, and its derivatives [29]. Among them, cellulose has received more and more attention HAp composite materials that include antibacterial compounds, such as chitosan or silver, can be produced using an in-situ precipitation method, or thermal spray technology. Caselis et. al. [30] prepared cotton cellulose modified with monochloroacetic acid and CaCl_2_ solution. This modification favored the growth of HAp on the surface of the cotton fibers in contact with simulated body fluid solutions (SBF).

Chitosan (Cs) is a cationic polysaccharide that is obtained from chitin deacetylation and is made of *N*-acetylglucosamine and glucosamine residues. It is an interesting biopolymer for medical applications due to the absence of toxicity, biocompatibility, biodegradability, anti-G(+), and G(−) bacteria activity, good film-forming ability, and promotion of cell proliferation [31]. Cs exerts its antibacterial activity by binding to the bacterial wall, entering into the cell, and forming a complex with DNA, thus inhibiting its replication [32]. It can also easily form chelate complexes with many antimicrobial agents [33]. Many researchers have tried to graft cellulose chains with Cs for biomedical and wound healing applications [34]. Cs-Ag ions complexes display lasting antibacterial activity against *Escherichia coli* and *Staphylococcus aureus* [35]. Cs has been used as naturally a reducing agent for the in-situ synthesis of Ag NPs on the surface of cotton gauze, which helps to reduce cytotoxicity, inflammatory, and accelerates wound healing. Moreover, Cs has been combined with HAp NPs to increase its bioactivity and osteoconductivity [36]. Yan et al. [37] coated a titanium substrate with a biocomposite containing Cs, Ag NPs, and HAp NPs and demonstrated that Cs-Ag-HAp NPs coatings can inhibit G(+) and G(−) bacterial growth through Ag and Cs synergistic effect, without any toxicity in osteoblastic MC3T3-E1 cells.

Nowadays, antimicrobial and UV protective medical textiles are mainly obtained by coating the cellulose fabric with functional NPs. Considering the importance of surface modification and insitu synthesis of NPs onto medical cotton gauze, the present study aimed to develop colored, antimicrobial, and UV-blocker cotton gauze fabrics by coating them with HAp NPs by wet precipitation of HAp NPs followed by Ag NP phytosynthesis using ginger oil as a reducing agent [38]. Chitosan was used in surface modification due to it is biodegradability, biocompatibility, and most importantly, promotes wound healing, features that make it suitable as a starting material for wound dressings. The cotton gauze samples were modified by the addition of Cs (cationic modification) or by partial carboxymethylation (anionic modification) to further improve HAp and Ag NPs dispersion and adhesion to the fiber surface. The HAp-Ag NPs coating provides good UV-protection and antimicrobial properties. The Ag NPs gave very high antimicrobial activities against different microorganisms (i.e., *Candida albicans*, *E. coli*, and *S. aureus*), and a brilliant yellow-brown coloration compared with the unmodified cotton gauze sample. The innovative strategy involved three distinct steps: (1) Cationization of cotton gauze by reacting it with chitosan or the anionization of cotton gauze through partial carboxymethylation using monochloroacetic acid. (2) Thus anionic and cationic modified cotton gauze along with unmodified samples (blank) was submitted to in situ formations of Ag NPs using ginger oil which has been used as a reducing agent and stabilizing agent to prevent aggregation of Ag NPs, and as a linker for fixation of Ag NPs on the surfaces of the cotton gauze. Furthermore, ginger oil has been chosen as it is a green reducing agent for Ag NPs synthesis and for treating inflammatory conditions and their associated pain. It is recognized as safe to use by the US Food and Drug Administration (FDA) with systemic and local (skin) anti-inflammatory effects [39].

## 2. Materials and Methods

### 2.1. Materials

Scoured and bleached weave cotton gauze samples were produced by the El-Nasr Spinning, Weaving, and Dyeing Company, Mahala, Egypt. Absolute ethanol (C_2_H_5_OH, 99%, ADWIC), glacial acetic acid (CH₃COOH, 99%, Sigma-Aldrich, St. Louis, MO, USA), phosphoric acid (H_3_PO_4_, 85 wt.% in H_2_O, Sigma-Aldrich), calcium hydroxide (Ca(OH)_2_, ≥96%, Sigma-Aldrich), and ammonium hydroxide (NH_4_OH, 28% NH_3_ in H_2_O, Sigma-Aldrich) were used for the synthesis of HAp NPs. Low molecular weight Cs (*M*_Wt_ = 30 KDa and 97% deacetylation, Acros Organics, Fair Lawn, NJ, USA), sodium hydroxide (NaOH, ≥97.0% pellets, Sigma-Aldrich), and monochloroacetic acid (C_2_H_3_ClO_2_, 99%, Sigma-Aldrich) were used for the cationic and anionic modification of the cotton gauze samples, respectively. Silver nitrate (AgNO_3_, 99%, Sigma-Aldrich), phenolphthalein (C_20_H1_4_O_4_, 98%, Merck, Kenilworth, NJ, USA), and natural ginger oil (Pure Grade, Sigma Aldrich) were used for Ag NPs phytosynthesis.

### 2.2. Cationic and Anionic Modification of the Cotton Gauze Samples

For the cationic modification, the Cs solution was prepared by adding 2 g of Cs to 100 mL acidic water (1% acetic acid) under constant overnight magnetic stirring. Then, cotton gauze samples (10 × 10 cm^2^) were put in the prepared Cs solution (liquor ratio = 1:30) and sonicated using the ultrasonic bath (A CREST Ultrasonic, TRU-SWEEPTM ultrasonic benchtop cleaner bath, model 575 D with a capacity 5.75 L) operating at 38.50 kHz at power level 40 W, 50 °C for 30 min, followed by drying at 90 °C for 15 min. The modified sample nitrogen content (percentage) was calculated using the semimicro method described by Kjeldhal with the Cole and Parks modification [40].

Anionic modification (A) was performed by partial carboxymethylation of cotton gauze samples in two steps. First, gauze samples (10 × 10 cm^2^) were impregnated with an aqueous sodium hydroxide solution (15%) at room temperature (RT) for 5 min, squeezed (wet pick-up = 100%), and then dried at 60 °C for 5 min. Second, samples were immersed in an aqueous solution of monochloroacetic acid 3 mol at RT for 5 min, followed by squeezing to 100% wet pick-up, and heating at 80 °C for 1 h, washing and drying at RT. The carboxyl content in the modified samples was quantified following the United States Pharmacopoeia method in which 0.5 g of sample was cut in short fibers that were immersed in 50 mL of 2 wt% calcium acetate solution for 15 h. Samples were then titrated with 0.1 M NaOH standard solution and phenolphthalein as an indicator. The carboxyl content was calculated with the following equation [41]:$$\mathrm{COOH}\;\%\;=\;\frac{N\;\times \;V\;\times \;M_{\mathrm{Wt}}\;\mathrm{COOH}}{m}$$
where *N* is the normality of the 0.1 M NaOH solution, *V* is the volume of consumed NaOH corrected for the unmodified cotton gauze (blank), *M*_Wt_ COOH is the carboxyl group molecular weight, and m is the weight of the oven-dried sample.

### 2.3. Wet Precipitation of Hydroxyapatite on the Cotton Gauze

Chemically unmodified (blank) and modified cotton gauze samples were immersed in 0.5 M calcium hydroxide solution (sparingly soluble) followed by drop-by-drop addition of 0.3 M phosphoric acid at 40 °C. The pH was adjusted to 8 by NH_4_OH addition during the precipitation. To study the effect of HAp content, the HAp NPs were deposited on the surface of cotton gauze at different concentrations (2.5 and 5 wt.% based on the dry weight of the unmodified cotton gauze). The HAp cotton gauze samples are as fellow: unmodified (H2.5 and H5), cationized (C-H2.5 and C-H5), and anionized (A-H2.5 and A-H5) cotton gauze samples.

### 2.4. Phytosynthesis of Ag NPs on Cotton Gauze Samples Using Ginger Oil

Unmodified (blank), cationized, and anionized cotton gauze samples (10 × 10 cm^2^) coated with HAp NPs were immersed in a silver nitrate solution (500 ppm) (liquor to fabric ratio = 1:50) at RT for 30 min, followed by the drop-by-drop addition of 10 mL of ginger oil in ethanol (10%) with vigorous stirring at 60 °C for 30 min. Then, samples were squeezed, rinsed in water, and dried at RT. The residual solution absorbance was measured by ultraviolet-visible (UV–vis) spectroscopy. To study the effect of HAp/Ag content, HAp/Ag NPs of different HAps concentrations (2.5 and 5 wt.%) were deposited on unmodified (H2.5-Ag and H5-Ag), cationized (C-H2.5-Ag and C-H5-Ag), and anionized (A-H2.5-Ag and A-H5-Ag) cotton gauze samples.

### 2.5. Morphological and Elemental Characterization

The prepared samples’ morphology and elemental composition were characterized using different analytical techniques. The cationic and anionic modifications of the sample surface were examined by recording by ATR technique using the Fourier transform-infrared (FTIR) spectra with a JASCO FT/IR-460 spectrophotometer (Japan) at the range of 400–4000 cm^−1^. The crystalline phase was identified with a D8 advance X-ray diffractometer (XRD, Bruker AXS D8, Berlin, Germany) and Cu-Kα radiation (*λ* = 0.154056 nm) and the following parameters: 2*θ* from 4° to 80° and 0.04° step, scanning speed of 0.4 s. Particle size, surface morphology, and elemental composition were examined using field emission scanning electron microscopy (FE-SEM) at accelerating voltage 20 kV and spatially resolved energy-dispersive X-ray spectroscopy (EDX, JEOL JSM-6510LV QSEM, Japan) using an advanced electron microscope with a LAB-6 cathode at 520 keV. Element maps of C, O, Ca, P, and Ag have been used to show the spatial distribution of Ag and HAp NPs on the cotton gauze. Applying higher voltages do not attain the ultimate high magnification without degrading the samples or changing the particle morphology. The HAp and Ag NP structure was analyzed by high-resolution transmission electron microscopy (HR-TEM) and acceleration voltage up to 200 kV (TEM, JEOL-JEM-2100, Tokyo, Japan). Thermal stability and ash contents (HAp and Ag NP content) in the modified cotton gauze samples were characterized by thermogravimetric analysis (TGA, Q5000 processor, TA Instruments, New Castle, DE, USA) under air atmosphere 20 mL/min and at heating–cooling rate of 10 °C min^−1^. The amount of Ag NPs (mg/kg) deposited on the cotton gauze samples was evaluated by atomic absorption spectrophotometer (AAS, contrAA^®^ 700, Analytik, Jena, Germany) using a silver lamp at 328.1 nm wavelength. The photoluminescence emission spectra were determined with a Spectro-fluorophotometer (Shimadzu RF-5301PC, Kyoto, Japan) to confirm Ag NPs deposition on the cotton gauze samples. The silver remaining in the solution was quantified by UV–vis spectroscopy (V-770 UV–Visible/NIR spectrophotometer, JASCO, Portland, OR, USA) over a range of 200 to 800 nm, and the results were compared with those obtained by AAS.

### 2.6. Zeta Potential, Ultraviolet Protection, and Colorimetric Features Tests

Enhancement of functional properties may be explained by modifications of surface properties, particularly surface energy and electrical parameters at the surface, measured by zeta potential [42]. The zeta potential of cotton gauze was measured employing equipment provided by the Anton Paar Austria GmbH. Evaluating UV-protection activity for fabrics by AATCC 183:2010. This standard test method is used to determine the UV-radiation blocked or transmitted by cotton gauze intended to be used for UV protection [43]. The samples’ Ultraviolet Protection Factor (UPF) was determined using the UPF calculation system of a UV–Vis spectrophotometer (AATCC test method 183:2010 for UVA Transmittance evaluation). The colorimetric properties of the cotton gauze samples were determined using the CIELAB color space values (L*, a*, b*) and a Hunter Lab spectrophotometer (UltraScan-Pro, USA). L* a*, and b* are the lightness from white (100) to black (0), from red (+) to green (−), and from yellow (+) to blue (−), respectively.

### 2.7. Antimicrobial Evaluation Tests

The antimicrobial activities were investigated with the agar disc diffusion technique. Briefly, the G+ *S. aureus* (ATCC 6538), the G− *E. coli* (ATCC 25922), and the fungi *Candida albicans* (ATCC 10231) and *Aspergillus niger* (NRRL A-326) were inoculated (0.1 mL of 10^5^–10^6^ cells/mL) in nutrient agar plates. Then, unmodified and modified gauze samples were put on top of the agar, and plates were left at 4 °C for 2–4 h to allow dispersion of the antimicrobial compounds. Plates were then moved to 37 °C for 24 h for bacteria, and to 30 °C for 48 h for fungi. The sample antimicrobial activity was determined by measuring the diameter of the growth inhibition zone (in mm). Data were the mean values of several independent experiments [44].

### 2.8. Air permeability, Water Absorbance, and Tensile Strength Measurements

The comfort of unmodified and modified cotton gauze samples was assessed by investigating their (i) tensile strength (stiffness) with an Instron universal testing instrument (Model 4206, Instron Ltd., Norwood, MA, USA) according to the ASTM D 882-12 standard method, (ii) air permeability, according to the ASTM D737-99 standard method, and (iii) water absorbance according to ASTM D 4772 standard test.

### 2.9. Statistical Analysis

Air permeability, water absorption, and tensile strength were triplicated, and the net averages were measured. The results were expressed as the mean standard error and calculated using Microsoft Office 2010, Excel Program.

## 3. Results and Discussions

### 3.1. Chemical Modification Mechanisms

First, the surface properties of the prepared gauze samples were determined by calculating the zeta potential [45]. The zeta potential of cellulosic fabrics and fibers was low (from −20.6 to −24.5 mV) due to the presence of hydroxyl (–OH) and carboxyl (–COOH) groups. In addition to these groups, the swelling properties of the cellulosic fibers also contribute to the zeta potential value because swelling increases the surface area. This leads to the shift of the shear plane into the liquid phase, thus slightly decreasing the zeta potential [45].

### 3.2. Cationic and Anionic Modification Mechanisms

Cotton gauze samples underwent cationic (Cs-NH_2_) and anionic (partial carboxymethylation-COO^−^) modification to provide the targeted multifunctional properties (Figure 1). Cotton gauze samples were coated with cationic Cs-NH_2_ via H-bonds between Cs amino (–NH_2_) groups and the cellulose fiber hydroxyl groups (–OH). Carboxymethyl groups were introduced on cotton gauze via a 2-step partial carboxymethylation reaction: (i) partial conversion of the sample hydroxyl groups (–OH) to sodium salt (–ONa) by soaking in the NaOH solution; and (ii) etherification via nucleophilic substitution between chloroacetic acid and sodium cellulose. This is a non-specific reaction because it can occur at –OH groups located at the 2, 3, or 6 positions in the cellulose anhydrous glucose unit. After cationic modification, cotton gauze fabrics have more positively charged groups (NH_3_^+^). Conversely, the partially carboxymethylated cotton gauze fabrics (an ionization) have more negatively charged groups (methyl carboxyl groups –COO^−^). Measurement of the amino group (–NH_2_) and carboxyl group (–COOH) concentration in the samples confirmed the chemical modification in cationized (nitrogen content % = 0.32) and partially carboxymethylated cotton gauze fabrics (degree of substitution = 0.71).

The chemical bonding of the cationic Cs and anionic monochloroacetic acid (MCA) modifiers were studied by FTIR spectroscopy (Figure 2). The FTIR spectra of unmodified cotton gauze (blank) (Figure 2) included a broad peak from 3000 to 3600 cm^−1^ that corresponds to O–H stretching (Figure 2a). The intensity of this peak was reduced in Cs-treated cotton gauze samples and was explained by –OH stretching vibration of cellulose and Cs at 3400 cm^−1^, and by C–H stretching at 3000–2800 cm^−1^. Despite the presence of –CH_2_– groups in the cellulose structure, the peaks due to the symmetric and asymmetric stretching modes could not be individualized (Figure 2a). The peak at ~1640 cm^−1^ corresponded to the adsorbed water molecules. The cationic modification led to an absorption peak at 3400 cm^−1^ due to Cs amino groups. The peak at 1158 cm^−1^ corresponded to C–O–C stretching. After anionic modification with MCA, broad peaks were detected at 1700 cm^−1,^ and the peak intensity increased with a peak shift at 1600 cm^−1^, due to the presence of carboxylate anions (–COO^−^) (Figure 2b). The peak at 1060 cm^−1^, attributed to C–O–C saturated ether stretching, confirmed the successful anchoring. Compared with the unmodified cotton gauze (blank), the different intensity of the peak at 1317 cm^−1^ in Cs-modified samples could be explained by C–N stretching of Cs, and the peak at 1600–1500 cm^−1^ by N–H bending upon Cs incorporation.

In the Appendix A shows the FTIR spectra of pure MCA and Cs powders for comparison. The strong brand in the region 3291–3361 cm^−1^ corresponds to N–H and O–H stretching, as well as the intramolecular hydrogen bonds. The absorption bands at around 2921 and 2877 cm^−1^ are attributed to C–H symmetric and asymmetric stretching. The presence of residual N-acetyl groups was confirmed by the bands at around 1645 cm^−1^ (C=O stretching of amide I) and 1325 cm^−1^ (C–N stretching of amide III), respectively. The small band at 1550 cm^−1^ corresponds to the N–H bending of amide II. A peak at 1589 cm^−1^ corresponds to the N–H bending of the primary amine. The –CH_2_ bending and –CH_3_ symmetrical deformations were confirmed by the presence of bands at around 1423 and 1375 cm^−1^, respectively. The absorption band at 1153 cm^−1^ can be attributed to asymmetric stretching of the C–O–C bridge. The bands at 1066 and 1028 cm^−1^ correspond to C–O stretching. Appendix A shows the cotton-Ag and cotton-H5-Ag Raman. The obtained peaks are related to the cellulose chain of the cotton gauze. Cotton gauze samples (cotton-Ag and cotton-H5-Ag) showed strong, well-resolved peaks corresponding to ν of the C–C ring asymmetric stretching and C–O–C for glycoside link asymmetric and symmetric stretching at 1116,1331 and 1088 cm^−1^, respectively. The other peaks obtained at 1478, 379, and 1292 cm^−1^ are attributed to different types of –CH_2_ group vibrations, while peaks at 1337 and 995 cm^−1^ are assigned to C–OH groups as shown in Appendix A. The peak at 2897 cm^−1^ is assigned to CH and –CH_2_ stretching in cellulose. While peak obtained at 2736 cm^−1^ is attributed to the methine group in cotton. The spectrum of cotton-Ag and cotton-H5-Ag were similar. A new peak appeared at 540 cm^−1^ is assigned to HAp. 

### 3.3. Hydroxyapatite Wet-Precipitation Mechanism

HAp NPs were deposited on cotton gauze samples by wet in-situ precipitation. The zeta potential values of HAp suspensions indicated that stoichiometric HAp (bulk Ca–P ratio = 1.67) has a negative charge. Fahami et al. studied the surface charges of HAp particles and found negative zeta potentials at basic pH = 9 (−35.5 mV) [46]. This may be due to the high concentration of PO_4_^3−^ groups in the first few nanometers of the NP surface [47]. The Ca^2+^-deficient surface layer (Ca–P ratio = 1.67) is explained by the solid–solution equilibrium during HAp precipitation, leading to the creation of a vacancy on one of the ten Ca^2+^ sites and one of the two –OH sites, and the protonation of one of the six PO_4_^3−^ groups [47]. Cotton gauze surface coating with HAp NPs involved the reaction between H_3_PO_4_ and Ca(OH)_2_ (cotton gauze-HAp NPs) (Figure 3). Typically, the HAp wet-precipitation reaction (Equation (1)) starts with an ionic exchange between Ca^2+^ and H^+^ of the cellulose fiber hydroxyl groups (Equation (2)). Then, Ca^+^ ions interact with (HPO_4_)^2−^ groups forming HAp clusters of amorphous HAp onto the cotton fiber surface.
10 [Ca(OH)_2_] + 6 H_3_PO_4_ → [Ca_10_(PO_4_)_6_(OH)_2_] + 18 H_2_O(1)
(Cotton)–OH + Ca^2+^ → (Cotton)–O-Ca^+^ + H^+^(2)
(Cotton)–O-Ca^+^ + (HPO_4_)^2−^ → (Cotton)–O-Ca-OPOOHO(3)

Cationization of cotton gauze samples with Cs led to some shrinkage and to smaller holes in the cotton fabrics, but this did not modify their permeability to water. As HAp has Ca–P ratio = 1.67 and a negative charge (as indicated by the zeta potential measurements of HAp suspensions), the cationic (Cs) modification increased the cotton gauze adsorption capacities for HAp NPs compared with the anionic modification (MCA). However, the anionic modification (MCA) improved HAp NPs deposition compared with the unmodified sample. This could be explained by the carboxyl group’s ability to attract more Ca^2+^ ions, thus enhancing HAp NP cluster formation and heterogeneous nucleation compared with the unmodified cotton gauze (blank), but to a lower extent compared with Cs-modified samples. These observations were validated by XRD, SEM, TEM, and TGA analysis.

### 3.4. Silver Nanoparticle Phytosynthesis Mechanism

Ag NP phytosynthesis on unmodified cotton gauze (blank), HAp NP-coated, cationized/HAp NP-coated, and anionized/HAp NP-coated cotton gauze were performed using ginger oil as an eco-friendly reducing agent. Ginger oil main constituents (i.e., sesquiterpenes, farnesene, methylheptenone, cineol, borneol, geraniol, and linalool) (Figure 4) can exert reducing and stabilizing (capping) functions during Ag NP synthesis. Sesquiterpenes, farnesene, and methylheptenone contain π electrons, whereas borneol, geraniol, and linalool contain –OH groups [39]. The π electrons and –OH groups promote Ag^+^ phytoreduction to Ag^0^ and stabilize Ag NPs via the concomitant binding to the newly formed Ag NPs inside the cotton gauze fabrics. The π electrons can transfer to Ag^+^ ions free orbital and convert them to free Ag^0^. Ag^+^ ions can also be reduced to Ag NPs by an electron from the –OH groups of borneol, geraniol, and linalool, leaving these compounds in the quinone form [21].

Ag NPs were in situ deposited on the different gauze samples in three steps: pre-nucleation, nucleation, and growth. In the first step, cotton gauze samples have a negative zeta potential in the neutral solution due to the presence of –COOH or –OH groups in their chemical structure (Figure 4). In the AgNO_3_ solution, the electrostatic interaction between Ag^+^ ions and negatively charged groups (–OH, COOH) in the cellulose chain leads to Ag^+^ adsorption and diffusion on the gauze surface (Figure 4). The electrostatic interaction forces might reduce Ag^+^ mobility, thus promoting the heterogenous formation (step 1) of Ag clusters (i.e., the aggregation of a group of atoms/ions), and controlling their development. During nucleation (step 2), Ag^+^ ions on the sample surfaces undergo reduction by bioactive molecules found in ginger oil and in the cellulose chain. In the third step (growth), silver clusters agglomerate to form nuclei (seed crystals) with a critical size that can group to form stable Ag NPs. More information on Ag NP formation can be found in the literature [48].

### 3.5. Crystalline Structure of HAPs/Ag/Cotton Gauze

The XRD patterns of unmodified cotton gauze (blank) and HAp/Ag NP-coated cotton gauze samples (Figure 5a–c) highlighted the presence of diffraction peaks at 2*θ* = 14.8° and at 2*θ* ~16.5°, 22.5°, and 34.5° that corresponded to the cellulose I (110), (200), and (004) crystalline lattice planes. The peak intensity was not significantly influenced by cationization (with Cs). The relative crystallinity values were 58.3 and 59.7% for unmodified cotton gauze (blank) and Cs-modified samples, indicating that cationization (Cs) did not modify the sample’s main crystalline form and crystal structure. The sharp peak with high intensity at an angle = 25.9° and 31.8° with miller indices (*hkl* values) of (002) and (211) corresponded to HAp crystals. The XRD analysis (Figure 5a) indicated that HAp crystals had a hexagonal structure (PDF card number 009-0432 and 01-072-1243). HAp crystals were made of small crystallites of about 5 nm (the Scherrer formula was used to calculate the crystallite size). The diffraction peaks at ~16.5° and 22.8° progressively disappeared with the higher HAp concentration. The peak at 2*θ* = 47.1° corresponded to HAp (222) lattice planes [26,49,50]. Unlike HAp NPs, Ag NPs could not be detected possibly because of their low content or small crystallite size (high amorphous content).

Cotton gauze samples modified with Cs and coated with HAp and Ag NPs (C-H2.5-Ag and C-H5-Ag) showed the highest peak intensity at 22.8° (Figure 5b) compared with the unmodified cotton gauze (blank) and the samples with anionic modification (carboxymethylation) (H2.5-Ag, H5-Ag) in which the main peaks also were slightly shifted to higher lower angles. A slight shift was observed for the A-H2.5-Ag sample (cotton gauze with anionic modification and coated with HAp and Ag NPs). Similarly, the samples modified with Cs and coated with HAp and Ag NPs (C-H2.5-Ag and C-H5-Ag) displayed the highest intensity for the peaks corresponding to the cellulose crystalline structure at 14.8° and 16.3° (Figure 5a).

### 3.6. Morphological, Structural, and Elemental Analyses

SEM analysis confirmed HAp/Ag NPs formation onto the sample surface. Comparison by SEM of the surface morphology of the different samples (blank, Ag, H2.5-Ag, H5-Ag, C-H5-Ag, M-H5-Ag) to determine the effects of HAp concentration (2.5 and 5 wt%) and the cationic and anionic modifications (Figure 6) showed that the unmodified cotton gauze (blank) had a smooth surface (Figure 6). Conversely, Ag and HAp-Ag NPs were detected on the surface of the H2.5-Ag, H5-Ag, C-H5-Ag, M-H5-Ag samples. HAp-Ag NPs amount progressively increased from unmodified cotton gauze (blank), cotton gauze with the anionic (carboxyl group) modification to cotton gauze with the cationic (Cs) modification. Moreover, HAp-Ag NPs content was higher in H5-Ag than H2.5-Ag samples mainly due to the HAp concentration increase from 2.5 to 5 wt%. The NPs composition was investigated by EDX analysis.

Elemental composition and HAp and Ag NPs distribution on the surface of the different samples were studied by SEM-EDX with elemental mapping (Figure 7). The Ag-coated cotton gauze was made mainly of C, O, and Ag, and HAP-Ag-coated cotton gauze was composed of C, O, Ca, P, and Ag. The data extracted from SEM-EDX analyses (Appendix A) showed areas of higher HAp NP concertation (Ca and P) in the cationized samples (Cs) than in the anionized samples and in the unmodified cotton gauze (blank). As a result of the HAp NPs (negatively charged) coating, the deposition of the Ag NPs on the chitosan modified (C-HAp5) cotton gauze will increase compared with the unmodified cotton gauze. Due to the cleating functional groups (–NH_2_ and –OH) in its structure, Cs can work as an excellent chelating agent and a stabilizer, as well as a reducing agent for the formation of Ag NPs and protecting them from agglomeration. In the case of anionic modification (carboxymethylation), the cotton gauze the –OH groups on the cellulose chain will be changed to –CH_2_–COOH group. The anionic modification will enhance the deposition of the Ag NPs on the cotton gauze. Therefore, more likely that both the anionic and cationic modification, as well as deposition of the HAp NPs, will increase the Ag NPs deposition on the cotton gauze. However, the distribution of the Ag NPs might be better in the case of the anionic modification (carboxymethylation) as shown by the SEM-elemental mapping. TEM was used to more accurately determine the size and morphology of the synthesized HAp and Ag NPs (Figure 8). Ag NPs had a spherical morphology with a mean particle size of 14 nm (range: 8–26 nm) (Figure 8a). HAp NPs had a rod-like shape and mean particle diameter and length of 20 nm and 138 nm, respectively (Figure 8b). As already shown by SEM, the TEM images (Figure 8c) confirmed Ag NPs deposition onto HAp-coated cotton gauze samples. However, these particles were weakly bound, and large agglomerates could be easily broken by ultrasonic agitation using an ultrasonic bath at power 40 W, 50 °C for 30 min in an aqueous solution, as done for the TEM samples.

### 3.7. Thermal Stability and Ash Content

The TGA data for the HAp and Ag NP-coated samples (Appendix A) showed a limited weight loss (~4 wt. %) between 70 and 150 °C caused by the loss of physically bound water after the main thermal decomposition steps (Table 1). The main decomposition step of the cotton gauze samples (~75 wt. %) showed an onset decomposition temperature of 296–320 °C and a maximum decomposition rate from 324–350 °C. This peak could be explained by the cellulose chain cleavage into smaller chains and the liberation of CO_2_ and H_2_O.

The TGA data for the Ag NP-coated samples were slightly different from those of the unmodified cotton gauze (blank). A second decomposition step was observed at 440 °C only for the samples modified with Ag NPs (~15 wt.%) and was caused by the cellulose chain complete thermal oxidation. In-situ Ag NP and HAp NP incorporation in the samples reduced their thermal stability. In conclusion, Ag and HAp NP incorporation reduced intermolecular interactions and thermal stability and increased the oxidation degree of the cellulose chains compared with Ag NPs alone. The residual ash content at 700 °C is listed in Table 1. The TGA and SEM-EDX analyses (Table 1 and Appendix A) confirmed that compared with the unmodified cotton gauze (blank), the cationic (Cs) modification increased HAp NP deposition on the cotton gauze, while the anionic modification (carboxylation or deposition of HAps NPs) mainly increases Ag NP deposition than the unmodified cotton gauze.

### 3.8. Silver Content and UV Protection

Upon addition of ginger oil in the AgNO_3_ solution containing the cotton gauze samples, the solution and sample color changed from yellow to yellowish-brown. This color change was caused by increased surface plasmon resonance of the formed Ag NPs. UV–vis spectrophotometry confirmed that: (i) ginger oil efficiently reduced Ag^+^ ions; and (ii) AgNO_3_ was completely reduced into Ag NPs in 30 min. In this section, Ag NP phytosynthesis using ginger oil, the UV protection properties of Ag-HAp nanocomposites, and the cationic and anionic modification effect on the amount of the deposited Ag NPs (determined by UV–vis spectroscopy and photoluminescence spectroscopy) will be discussed more in detail. The UV–Vis absorption spectra (Figure 9) showed an absorption peak in the yellow color wavelength range (400–450 nm) that was related to surface plasmon resonance, confirming Ag NP formation. The shoulder peak at 348 nm, which is commonly observed for Ag, was related to the plasmon resonance of bulk Ag NPs. Ag NP peak was more intense (higher) in anionized (A-H5-Ag) and cationized (C-H5-Ag) cotton gauze that the unmodified samples (H2.5-Ag and H5-Ag). This indicates that both anionic and cationic modification better promotes Ag NPs deposition than unmodified cotton gauze.

To confirm these results, the charge separation and recombination of the photogenerated charge carriers (e^−^/h^+^ pairs) was assessed by photoluminescence spectroscopy in the synthesized HAp-Ag NPs particles (Figure 5). HAp-Ag NPs were excited at 310 nm and emitted at ~445 nm at RT. However, the peak intensity of HAp-Ag NPs was greatly reduced due to the lower charge recombination, possibly indicating efficient resonance and charge separation in the visible region. The decreased emission intensity led to a reduction in the electron-hole recombination rate, and consequently to higher oxidation capacity of the synthesized HAp-Ag NPs. The charge carrier recombination progressively decreased and the lowest values were observed in the A-H5-Ag sample (high Ag NP contents with good distribution over the surface). This is in line with the SEM-EDS data and will be confirmed by the antimicrobial activity tests (better performance for the cationic and anionic modified cotton gauze) in the next sections.

The cationic and anionic modifications strongly influenced Ag NPs incorporation in the cotton gauze and their coloration (Table 2). Specifically, (i) Ag content was higher and active groups more numerous in the modified cotton gauze than in the unmodified cotton gauze (blank). The higher number of active –NH_2_ or –COO^−^ groups could promote Ag^+^ absorption on the sample surface, and consequently Ag^+^ conversion into Ag NPs, ultimately leading to higher Ag content and darker color. (ii) The samples that underwent anionization (carboxymethylation) and HAp NPs coating had the high Ag NPs content with better Ag NPs distribution of the surface. (iii) Unmodified cotton gauze (Blank) had lower Ag NP content than cotton gauze coated with HAp NPs without cationic or anionic modification. In the case of anionic modification, the HAp NPs have a negative potential and the presence of the carboxyl groups (–COO^−^) added more reactive groups in the cotton gauze. Therefore, more likely Ag^+^ ions can be adsorbed with a better distribution on the surface and bulk of anionized cotton gauze than cationic modification. (iv) Ag NP content was not significantly different in cotton gauze coated with different HAp contents (2.5 and 5%) and with neither cationic nor anionic modification.

Human skin is mainly sensitive to UV-A (320–400 nm) that can cause different types of damage (e.g., freckles, sunburn, and skin cancer). The main role of UV-protective clothing is to protect skin against the harmful effects of the sun. Surface modification of textiles against UV radiation especially for medical textiles becomes more important. The UV protection properties of the different samples were determined by calculating their UPF (Table 2). Unmodified and HAp-modified cotton gauze displayed low UPF values that were strongly increased after the Ag NP in-situ deposition. The MCA/HAp NP (2.5%)/Ag NP sample had the highest UPF value. This could be explained by the fabric’s high UV absorption due to the color change, with good UV scattering due to Ag NP’s high refractive index.

### 3.9. Colorimetric Properties

Upon Ag NP incorporation, the cotton gauze samples displayed a yellow-brownish color. The colorimetric analysis (Table 2) showed that the unmodified cotton gauze (blank) was white: L* = 88.6, a* = −0.02), and b* = 1.2. The color of HAp-modified samples (without Ag NPs) was not different from that of the unmodified cotton gauze (blank), as indicated by their very similar L*, a*, and b* values. Conversely, the L*, a*, b* values of the unmodified and Ag-modified cotton gauze were significantly different from those of the samples without Ag. Specifically, the L* values decreased due to the fabric change in color. The a* value increase indicated the presence of redness in the samples, whereas the b* value increase was explained by the yellow or golden yellow color upon Ag NP deposition. In all treated samples, the a* and b* values were positive, indicating that the samples were detected as reddish and yellowish-brown. The A-H-2.5-Ag and A-H5-Ag samples had the lowest L* (61.4–61.6) and the highest a* and b* values (7.3–7.1 and 19.5–19.2), possibly due to their high Ag NP content. Anisotropic Ag NPs exhibit a variety of brilliant colors because of their surface plasmon resonance (SPR) [51]. The anisotropic Ag NPs cannot be only used to color cotton gauze with beautiful and variable colors but they can also render the dyed fabrics antibacterial. Wu et al. [51] demonstrated a solution-dipping method for the fabrication of fluorinated decyl polyhedral oligomeric silsesquioxane (F-POSS)/AgNPs/branched poly(ethylenimine)-coated cotton fabrics with tunable colors (i.e., yellow, orange, red, blue, green, and violet) as well as durable antibacterial and selfhealing superhydrophobic properties.

### 3.10. Antimicrobial Activities

The development of fabrics with protective features is crucial for many applications, particularly in the health sector. The antimicrobial activities of the different cotton gauze samples were tested using *S. aureus* (G+), *E. coli* (G−), and the fungi *C. Albicans* and *A. niger* and the qualitative (inhibition zone) method (Appendix A). The unmodified and Ag-modified cotton gauze without Ag NPs did not show any antimicrobial activity against the tested microorganisms, as indicated by the absence of the inhibition zone. Conversely, Ag NP incorporation in the samples led to a good antimicrobial performance (i.e., presence of the inhibition zone) against *E. coli* (G−) (highest effect), *S. aureus* (G+), and *C. Albicans*, but not *A. niger* (Figure 10). The A-HAp2.5-Ag sample showed the highest effect (i.e., largest inhibition zones) against *S. aureus* (18 nm), *E. coli* (20 nm), and *C. Albicans* (18 nm).

Recently, increasing attention has been attracted by HAp doped with Ag^+^ ions (Ag-HAp). The Ag-HAp exhibits strong antibacterial activity as the bacterial cells are attracted by the electrostatic force to the surface of the HAp, where there is a direct interaction between the bacterial cell membrane and Ag^+^ ions [52]. Ciobanu et al. [53] studied the antibacterial activity of Ag-doped HAp NPs against G-(+) and G-(−) bacteria. The results showed that antibacterial activity increased with the increase in x-Ag (from 0.05 to 0.3) in the samples. The Ag:HAp NPs concentration had little influence on the bacterial growth (*P. stuartii*). Lamkhao et al. [54] synthesized HAp powder with antibacterial properties using a microwave-assisted combustion method. The result of this study confirms that radicals were responsible for the antibacterial properties. Other studies reported Ag NP antimicrobial activity against different microorganisms. This property could have various explanations, but the main reason might be linked to Ag NP dissolution and Ag^+^ release. Ag^+^ ions can interact with the negatively charged microbe membrane through electrostatic attractions, thus hindering their growth. Moreover, Ag^+^ ions can react with the –SH groups of microbe enzymes and inhibit them. Ag NPs can also enter inside the microorganisms and cause DNA damage. Our results indicated that Ag NPs incorporated in the modified samples inhibited more efficiently G− than G+ bacteria. This is explained by the electrostatic attraction between the positively charged Ag^+^ ions and the negatively charged G− cell membranes that facilitates Ag^+^ ions attachment to the membrane. In fungi, Ag NPs act through depletion that forms irregularly shaped pits in the fungal outer membrane, and changes the membrane permeability, leading to the release of membrane lipopolysaccharides and proteins [55]. Figure 6 illustrates the possible mechanisms of Ag NPs antibacterial effects [56].

### 3.11. Air permeability, Water Absorbance, and Tensile Strength

The different modifications of the tested samples could influence their air permeability (i.e., the airflow rate through the fabric surface). Air permeability testing (Appendix A) showed a small reduction in the airflow due to the cationic and anionic modifications and the deposition of HAp (higher effect with the higher concentration) and Ag NPs. The air permeability percentages were lower in the HAp-Ag-Cotton gauze (249.2 ± 1.1 cm^3^/cm^2^/sec) than in the blank cotton gauze (255.3 ± 1.4 cm^3^/cm^2^/sec). However, airflow still provided enough oxygen for the healing process (about 5–6% decrease compared with the unmodified cotton gauze). Indeed, oxygen is a key factor for biomedical applications. Similarly, the hydrophilic character of cotton gauze fabrics is crucial for their efficiency as a biomedical product. In a similar study, Üreyen et al. [57] developed antibacterial functionalization of cotton fabrics with a finishing agent based on Ag-HAp powders. They found that the air permeability slightly decreased by 19.5%.

The water absorption percentages of the unmodified and modified cotton gauze are listed in Appendix A. The water absorption percentages were higher in the HAp-Ag-Cotton gauze (124 ± 1.4%) than in the blank cotton gauze (98 ± 1.4%), due to the effect of the HAp and Ag NP coating. It is well known that the hydrophilic nature of cotton gauze fabrics is due to their chemical structure with hydroxyl and carboxylic acid substituents that promote water absorption. Compared with the unmodified cotton gauze (blank), water absorbance was increased in the modified samples. This could be attributed to the presence of more hydrophilic compounds (e.g., Cs and carboxyl modification) and of HAp and Ag NPs on the cotton gauze surface that creates more hydrophilic groups. The mechanical property for the unmodified (blank) and modified cotton gauze was assessed by measuring their tensile strength (N/cm^2^) (Appendix A). Our results are in agreement with Abbasipour et al. [58] who treated cotton gauze chitosan/Ag/ZnO by the dip, dry, and cure method for achieving desired wound dressing properties. The chitosan/Ag/ZnO-modified cotton gauze showed an increase in water absorbency (38%) compared to the unmodified sample. Furthermore, their antibacterial efficiency was 99% for *Staphylococcus aureus* (*S. aureus*) and 96% for Escherichia coli (*E. Coli*). Hajimirzababa et al. [59] modified the cotton gauze with microcapsules of alginate and AgNPs. The samples showed antibacterial properties against *Staphylococcus aureus* (*S. aureus*) and 96% for *Escherichia coli* (*E. Coli*) with an increase in water absorbency (+55%), water holding capacity (+28%), and vertical wicking (33%) in comparison with the unmodified cotton gauze.

The tensile strength was higher in the HAp-Ag-Cotton gauze (540 ± 3 N/cm^2^) than in the blank cotton gauze (510 ± 3 N/cm^2^), due to the effect of the Cs and carboxymethyl modifications and HAp and Ag NP coating. These results showing that ultrasonic irradiations and experimental conditions did not damage cotton gauze structure to a significant level. His results agreed with Noman and Petrů [60] who modified the cotton samples with ZnO NPs using an in situ ultrasonic acoustic method and alkaline treatment. Aside from tensile properties, we have noticed slight shrinkage in the cotton gauze samples upon washing. Generally speaking, the cotton gauze fibers contain large amounts of -OH groups and they are highly hydrophilic [61]. Cellulose fibers in the cotton gauze exhibit a low degree of crystallization, so that when cotton gauze absorbs water, the bonding force among cellulose fibers (chains) is reduced markedly, which causes swelling. Therefore, when cotton fabrics are twisted and undergo plastic deformation. Consequently, the fabric shrinks and wrinkles [61].

## 4. Conclusions

Conventional properties of cotton gauze were greatly improved by depositing nanoparticles (NP) on the surface as a proper candidate for application as an advanced wound dress. In this study, HAp NPs and Ag NPs were used to impart multifunctional properties to cotton gauze fabrics, including brilliant color, UV protection, and antimicrobial activity, and anti-inflammation properties. An environmentally friendly reductant (ginger oil) was chosen for Ag NP phytosynthesis on cotton gauze fabrics. The cationic (Cs) and anionic (carboxylation) modifications have been improved HAp and Ag NP deposition on the cotton gauze, respectively. Importantly, Ag NPs deposition imparted brilliant colors, UV protection, and antimicrobial activity. Conversely, the cationic and anionic modifications and HAp NP deposition did not directly affect the cotton gauze color, UV protection, and antimicrobial features. However, they influenced Ag NP number and uniform distribution on the fabric surface, thus indirectly affecting these features. Ag NP incorporation by in situ phytosynthesis led to the highest antimicrobial activities against different microorganisms (i.e., *C. albicans*, *E. coli*, and *S. aureus*), improved the fabric UV protection, and imparted a brilliant yellow-brownish color. The as-prepared materials have potential biomedical applications in wound healing.

## Figures and Tables

**Figure 1 nanomaterials-11-00429-f001:**
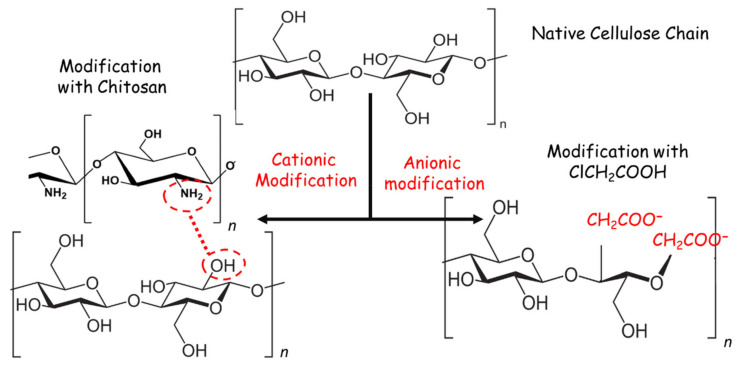
Schematic representation showing a surface modification of the cotton gauze using chitosan (Cs) (cationic modification) and monochloroacetic acid (MCA) (anionic modification).

**Figure 2 nanomaterials-11-00429-f002:**
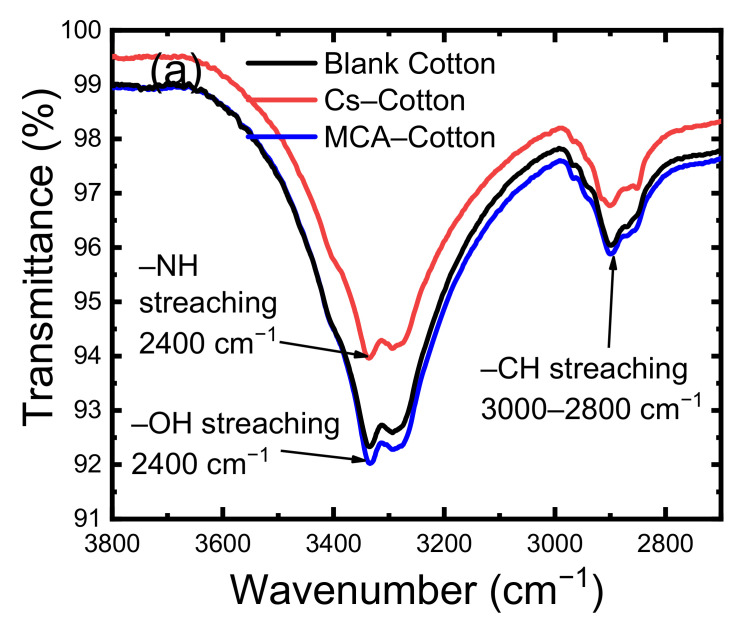
Fourier transform-infrared (FTIR) spectra of unmodified cotton gauze (blank), carboxymethylated cotton gauze (MCA-Cotton), and cotton gauze modified with cationic chitosan (Cs-Cotton): (**a**) spectra from 3800 to 2700 cm^−1^, (**b**) spectra from 1700 to 1200 cm^−1^, and (**c**) spectra from 1200 to 500 cm^−1^.

**Figure 3 nanomaterials-11-00429-f003:**
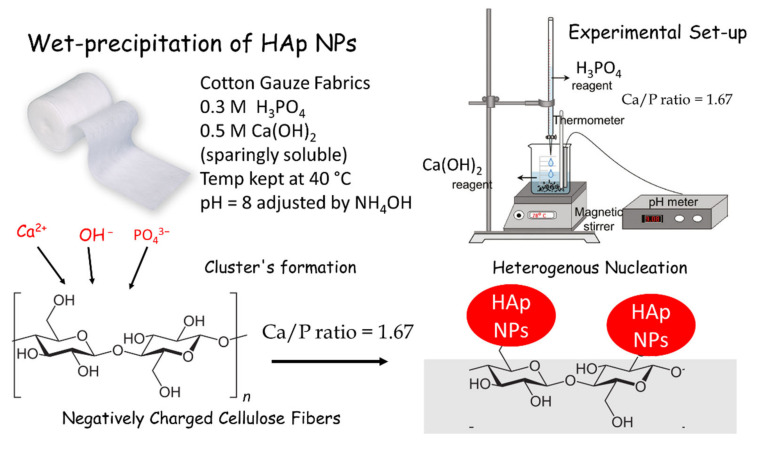
Schematic representation showing wet-precipitation of hydroxyapatite (HAp) and nanoparticles (NPs) on the surface of negatively charged cotton gauze fibers by primary adsorption or nucleation of the Ca^2+^/PO_4_^3−^/OH^−^ ions on the surface of cotton fibers.

**Figure 4 nanomaterials-11-00429-f004:**
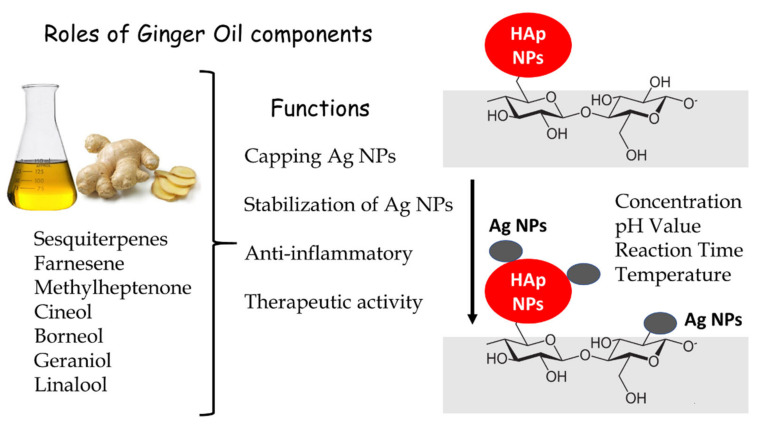
Schematic representation showing Ag NP phytosynthesis on the surface of HAp NP-coated cotton gauze in which the reducing agent was ginger oil, a green chemical with anti-inflammatory and therapeutic activities.

**Figure 5 nanomaterials-11-00429-f005:**
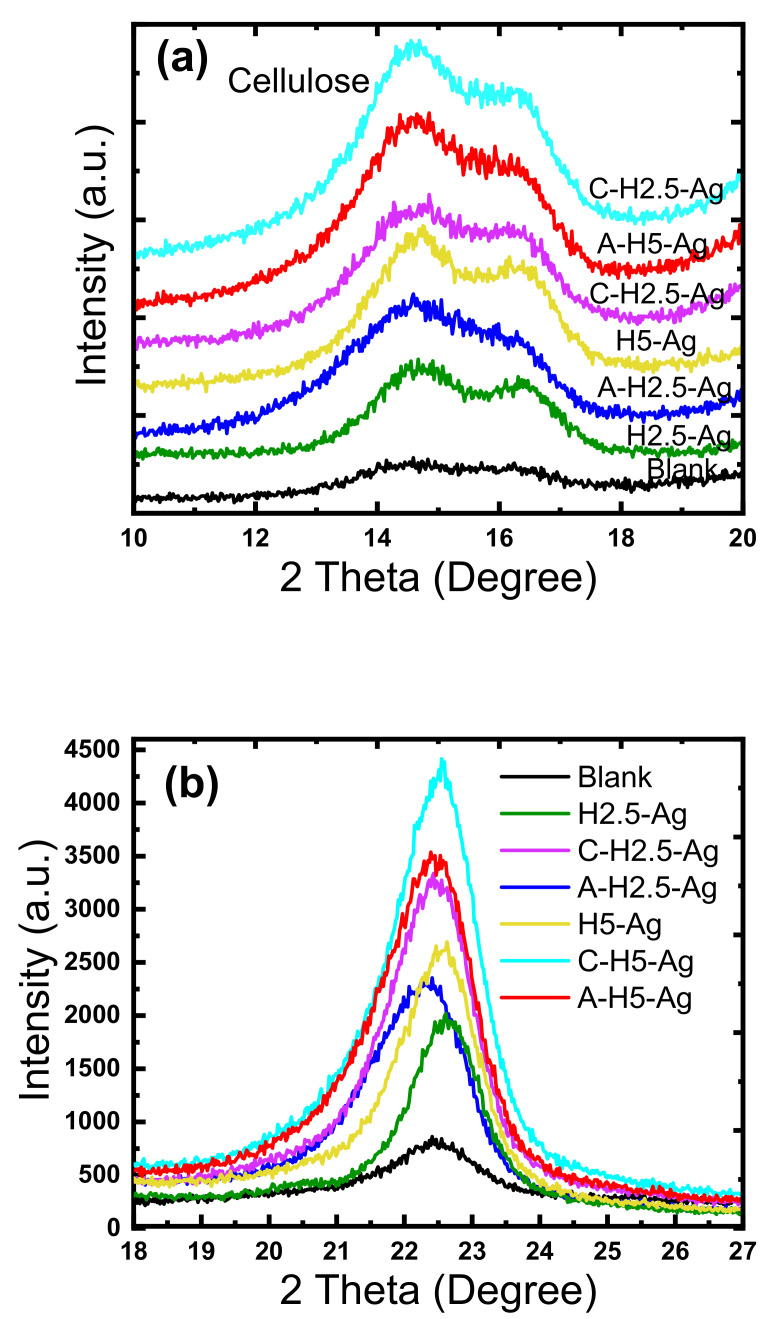
Characteristics of the X-ray diffraction (XRD) patterns of unmodified (blank) and modified cotton gauze samples at different 2 theta scale (**a**–**c**).

**Figure 6 nanomaterials-11-00429-f006:**
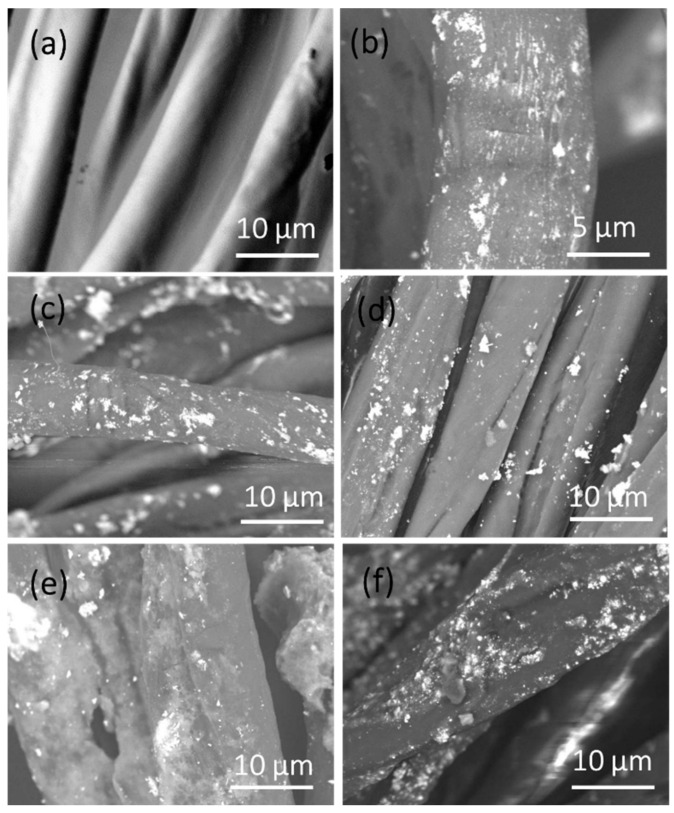
Representative scanning electron microscopy (SEM) photographs of the cotton gauze samples: (**a**) blank cotton gauze, (**b**) Ag- cotton gauze, (**c**) 2.5%HAp-Ag- cotton gauze, (**d**) 5%HAp-Ag-cotton gauze, (**e**) 5%HAp-cationized cotton gauze; (**f**) 5%HAp-anionic modified cotton gauze.

**Figure 7 nanomaterials-11-00429-f007:**
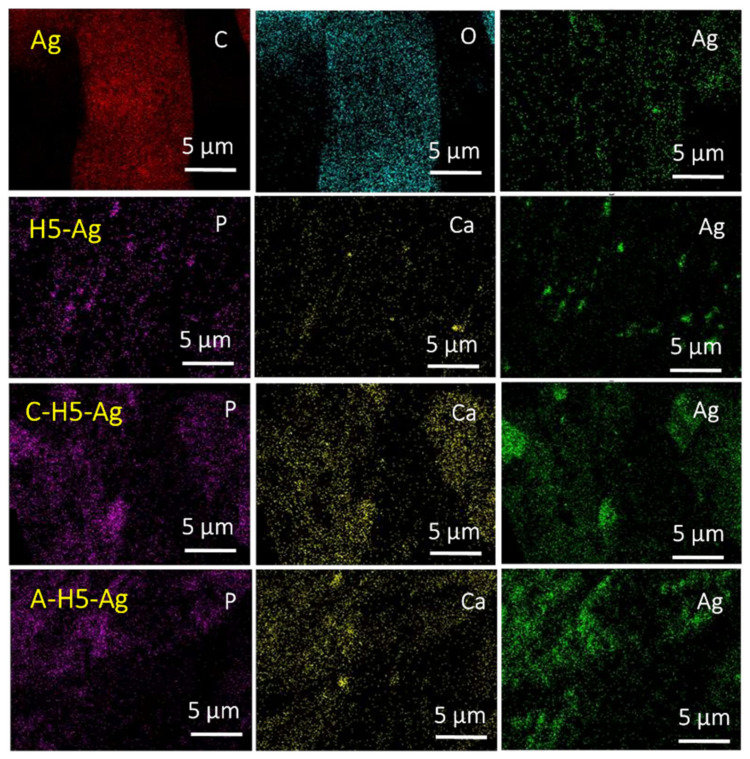
SEM-EDS analysis with the elemental mapping of cotton gauze samples coated with HAp-Ag NPs namely Ag-coated cotton gauze (Ag), Ag-HAp-coated cotton gauze (H5-Ag), cationized-HAp-Ag-cotton gauze (C-H5-Ag), and Anionic modified-HAp-Ag-cotton gauze.

**Figure 8 nanomaterials-11-00429-f008:**
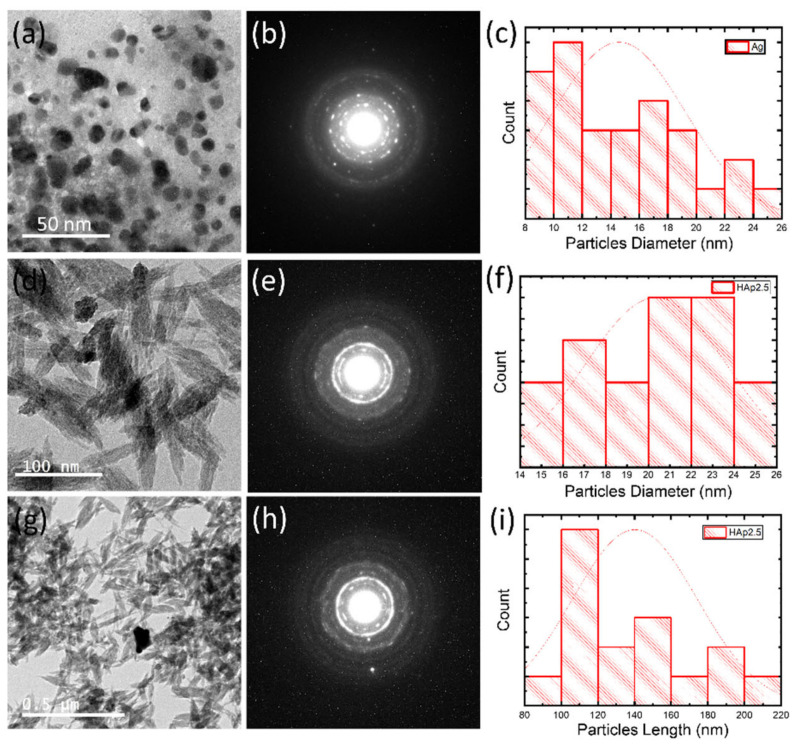
Representative transmission electron microscopy (TEM) images of the prepared nanoparticles and their corresponding selected area electron diffraction images and particle size distribution: (**a**–**c**) Ag NPs; (**d**–**f**) HAp2.5 NPs; (**g–i**) Ag-HAp2.5 NPs. In the histograms change into Particle Diameter.

**Figure 9 nanomaterials-11-00429-f009:**
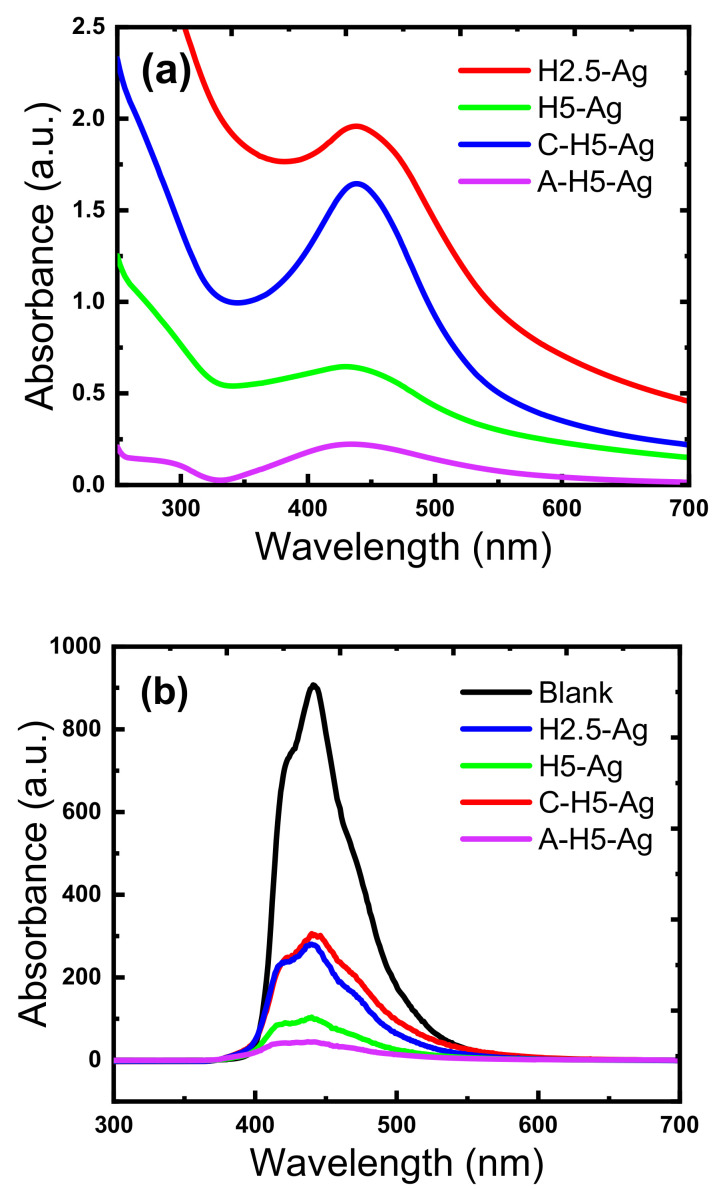
Spectroscopic characteristics of the blank, NPs modified (H2.5-Ag and H5-Ag), cationized (C-H2.5-Ag), and anionized (A-H5-Ag) cotton gauze samples: (**a**) UV–vis spectra of the remaining solution, and (**b**) photoluminescence spectra.

**Figure 10 nanomaterials-11-00429-f010:**
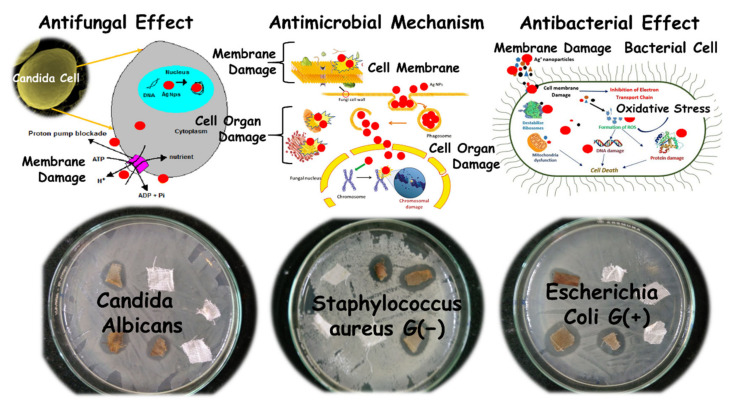
Ag NP antimicrobial effect in *Candida albicans*, *Escherichia coli* (G(+)), and *Staphylococcus aureus* (G(−)).

**Table 1 nanomaterials-11-00429-t001:** Data extracted from the thermogravimetric analysis (TGA) and SEM-EDS analyses with elemental analysis of the cotton gauze samples.

Sample	TGA Analysis			SEM-EDS Elemental Composition				
	*T* _onset_	*T* _max_	Ash	C	O	Ca	P	Ag
	°C	°C	%	%	%	%	%	%
Ag	327	350	1.5	47.8	43.8	–	–	8.4
H2.5-Ag	313	345	2.2	45.8	41.9	1.0	0.6	10.7
H5-Ag	296	325	3.7	45.7	35.8	1.5	1.1	15.8
C-H5-Ag	303	329	7.1	34.1	20.5	4.8	4.1	36.5
A-H5-Ag	298	324	6.2	35.3	29.2	3.6	1.8	30.1

**Table 2 nanomaterials-11-00429-t002:** Silver (absorption spectrophotometry (AAS)), Ultraviolet Protection Factor (UPF) (UV–vis spectrophotometry), and colorimetric analysis (According to Color Matching System CIE 1976 L*, a*, b* color space) of the indicated cotton gauze samples after washing and drying.

Samples	Silver Content	UPF	Colorimetric Analysis (CIELAB)
(mg/kg)	Factor	L*	a*	b*	RGB Color
Blank	0.0	8	88.6	−0.02	1.2	
Ag	3271	220	63.6	5.3	16.2	
H2.5	0.0	11	86.37	0.06	1.5	
C-H2.5	0.0	12	87.83	0.02	1.2	
A-H2.5	0.0	11	86.02	0.02	1.3	
H5	0.0	10	85.13	0.11	1.4	
C-H5	0.0	9	85.15	0.10	1.1	
A-H5	0.0	10	85.34	0.15	1.4	
H2.5-Ag	3714	235	62.4	5.6	16.8	
C-H2.5-Ag	3540	241	63.7	5.2	16.2	
A-H2.5-Ag	3846	246	61.4	7.3	19.5	
H5-Ag	3726	288	62.8	5.9	17.1	
C-H5-Ag	3631	240	63.3	5.1	16.3	
A-H5-Ag	3897	242	61.6	7.1	19.2	

## Data Availability

The data presented in this study are available on request from the corresponding author.

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
