# Peer review of "Multifunctional Hydroxyapatite/Silver Nanoparticles/Cotton Gauze for Antimicrobial and Biomedical Applications"

_nanomaterials, 2021, doi:10.3390/nano11020429_

Round 1
Reviewer 1 Report
The manuscript presented by Mohamed M. Said et al. concerning the fabrication of cotton gauze fabrics modified by in situ precipitation of hydroxyapatite nanoparticles and photosynthesis of silver nanoparticles with ginger oil for the medical approach is very interesting. The authors reported many characterizations and explained the characteristics of each sample in a proper way.
Even though the work is worthy of publication, there are some points that need to be clarified and information to add. The comments are given below:
Grammar form: The English form is good.
Title: Some words are improperly written with capital letters, please correct it.
According to the “Guidelines for Authors” of Nanomaterials the references through the text should be placed in square brackets [ ] and before the punctuation; for example [1], [1–3] or [1,3]. Please check all the references and correct according to the Guidelines.
Introduction
Line 39, 41-42: e.g. needs a comma at the end: “e.g.,”
Line 43: Since the medical textiles take place in the human body as the Authors said, I suggest changing “should be” with “must be”.
Line 59-66: The font size of this part seems to be different from the other parts of the text.
Line 129-132: This part could be classified more as a conclusion, thus, in my opinion, it seems out of context in the introduction.
Materials and Methods
Line 150: What is the instrument used for Cs solution sonication? It is an ultrasound bath or probe sonication? Please clarify and add the name and manufacturer of the respective instrument.
Line 150: The sentence “curing at 150 °C for 1 min” is not very clear, I suggest to Authors to explain the concept adding more details.
Lines 156: What the Authors with “wet pick-up 100%” refer to? Could they explain it better?
Lines 157: The sentence “in 3 mol of an aqueous solution of monochloroacetic acid sodium salt” is not written fluent, I suggest to modifying it as follow: “In aqueous solution of monochloroacetic acid sodium salt 3 mol”
Lines 158: The Authors reported the use of sodium salt of monochloroacetic acid for the second step of reaction; however, in the materials paragraph is mentioned the monochloroacetic acid and not its salt. Please, check and correct it.
Lines 163: The substance phenolphthalein is missing in the material paragraph. Please add it.
Equation line 164: what does the value “m” in the denominator refer to? Moreover, in the equation Authors write MWT but in line 166 they write MW to explain the value.
Lines 171-172: It is not clear how the HAp NPs at 2.5 wt.% and 5 wt.% were obtained. I strongly suggest explaining better this part.
Line 175: The “unmodified cotton gauze sample” prepared as a control is not clear if it is related to blank or cotton gauze with HAp NPs. I strongly suggest adding some information.
Line 190: Which elements will be analysed by Authors? They should be indicated in this paragraph, not just in the Results and Discussion section.
Line 208: I suggest to Authors adding a reference for UVA Transmittance evaluation test method.
Line 228: A paragraph regarding the statistic analyses is missing. I strongly suggest the Author to add it and use the statistical approach for all the data that require it.
Results and Discussion
Line 232: The method to determine the zeta potential of gauze is missing in the respective paragraph. Please add the information and the instrument used.
Line 254: The sentence “OK? Sentence not easy to understand” is out of context, please remove it.
Figure 1: Figure 1 should be divided into three separate figures. Each figure should be moved in the corresponding paragraph, that is, figure 1a in paragraph 3.1.1, figure 1b in paragraph 3.1.2 and figure 1c in paragraph 3.1.3.
Line 280: The supplementary figure related to FTIR spectra of pure MCA and Cs powders is the S1 and not the S2.
Line 287: Please add the characteristics of zeta potential analysis.
Line 302: How the Authors justify the sentence “this did not modify their permeability to water”? Do they have some data about water permeability of blank gauzes?
Lines 314-322: I think it is necessary adding some references for these statements.
Paragraph 3.2 This part is a little bit confuse. Firstly, the name of each samples (C-H2.5-Ag, C-H5-Ag, etc.) should be reported in paragraph “Method” the first time they are described in order to allow a better understanding of the text. The whole structure of the paragraph should be changed describing the samples in the same order they were prepared.
Line 370: The analysed sample (blank, Ag, H2.5-Ag, H5-370 Ag, C-H5-Ag, M-H5-Ag) should be also reported in the relative paragraph of “Materials and Methods” section. Moreover, the sample called “Ag” was not described in that section, thus it is not clear how it was obtained and its nature.
Lines 374-376: The SEM images demonstrate that the HAp/Ag NP are formed; however, the difference of HAp/Ag NP amounts onto the surfaces between the different samples, as the Authors affirm, is not clear. Could the Authors add other images, even in different magnification, in order to clarify it?
Lines 385-387: This sentence does not coincide with the images: “Elemental mapping by EDS showed areas of higher HAp NP (phosphorus and calcium) content in the cationized samples (Cs) than in the anionized samples and in the blank”. Elemental mapping reported by the Authors showed a difference in calcium content between cationized and blank sample, but there is no obvious difference with anionized one.
Table 1: According to the Authors statement the anionized sample has higher AgNps than cationized one but, in the table, the percentage of Ag in the A-H5-Ag sample is less than C-H5-Ag. These data are in contradiction. I strongly suggest to Authors to check the data and correct the table or the text.
Line 428: The figure 5 is already reported in the text, please correct it in figure 6.
Line 442: The figure 5c should be moved in the paragraph 3.5.
Line 450: The figure 5d should be moved in the paragraph 3.5.
Table 2: The table should be moved below the paragraph 3.6.
Author Response
Response to Reviewer 3
Grammar form: The English form is good.
Title: Some words are improperly written with capital letters, please correct it.
The title has been modified
According to the “Guidelines for Authors” of Nanomaterials the references through the text should be placed in square brackets [ ] and before the punctuation; for example [1], [1–3] or [1,3]. Please check all the references and correct them according to the Guidelines.
The references have been modified according to the journal guidelines.
Introduction
Line 39, 41-42: e.g. needs a comma at the end: “e.g.,”
The paper formate and minor mistakes have been corrected.
Line 43: Since the medical textiles take place in the human body as the Authors said, I suggest changing “should be” with “must be”.
The sentence has been corrected
Line 59-66: The font size of this part seems to be different from the other parts of the text.
The paper formate and minor mistakes have been corrected.
Line 129-132: This part could be classified more as a conclusion, thus, in my opinion, it seems out of context in the introduction.
The introduction (the last paragraph) has been revised.
Materials and Methods
Line 150: What is the instrument used for Cs solution sonication? It is an ultrasound bath or probe sonication? Please clarify and add the name and manufacturer of the respective instrument.
More information has been added to this section.
Line 150: The sentence “curing at 150 °C for 1 min” is not very clear, I suggest to the Authors to explain the concept by adding more details.
The sentence has been corrected
Lines 156: What the Authors with “wet pick-up 100%” refer to? Could they explain it better?
Wet pick-up is the amount of liquid absorbed by a fabric after it has been dipped and padded as a percentage of the weight of the dry fabric. For example, if the fabric weighs 10 oz/yd2 dry, and absorbs 10 oz/yd2 of liquid after being dipped and padded, weighing a total of 20 oz/yd2, the wet pickup is 100%.
Lines 157: The sentence “in 3 mol of an aqueous solution of monochloroacetic acid sodium salt” is not written fluent, I suggest to modifying it as follow: “In aqueous solution of monochloroacetic acid sodium salt 3 mol”
The sentence has been corrected.
Lines 158: The Authors reported the use of sodium salt of monochloroacetic acid for the second step of reaction; however, in the materials paragraph is mentioned the monochloroacetic acid and not its salt. Please, check and correct it.
The name has been changed to monochloroacetic acid, and it is the correct name.
Lines 163: The substance phenolphthalein is missing in the material paragraph. Please add it.
The substance has been added to the materials.
Equation line 164: what does the value “m” in the denominator refer to? Moreover, in the equation Authors write MWT but in line 166 they write MW to explain the value.
The sentence has been corrected.
Lines 171-172: It is not clear how the HAp NPs at 2.5 wt.% and 5 wt.% were obtained. I strongly suggest explaining better this part.
This sentence has been highlighted in the experimental section: To study the effect of HAp content, HAp NPs of different concentrations (2.5 wt.% and 5 wt.%) were deposited on unmodified, cationized, and anionized cotton gauze samples.
Line 175: The “unmodified cotton gauze sample” prepared as a control is not clear if it is related to blank or cotton gauze with HAp NPs. I strongly suggest adding some information.
The unmodified cotton gauze (blank) was clarified in the full manuscript.
Line 190: Which elements will be analyzed by the Authors? They should be indicated in this paragraph, not just in the Results and Discussion section.
Element maps of C, O, Ca, P, and Ag has been used to show the spatial distribution of Ag and HAp NPs on the cotton gauze
Line 208: I suggest to the Authors adding a reference for UVA Transmittance evaluation test method.
The section has been modified and ref has been added.
Line 228: A paragraph regarding the statistic analyses is missing. I strongly suggest the Author to add it and use the statistical approach for all the data that require it.
2.9. Statistical analysis
Air permeability, water absorption, and tensile strength were triplicated, and the net averages were measured. The results were expressed as the mean standard error and calculated using Microsoft Excel Program (2010).
Results and Discussion
Line 232: The method to determine the zeta potential of gauze is missing in the respective paragraph. Please add the information and the instrument used.
More information has been added to the experimental section.
Line 254: The sentence “OK? Sentence not easy to understand” is out of context, please remove it.
The sentence has been deleted.
Figure 1: Figure 1 should be divided into three separate figures. Each figure should be moved in the corresponding paragraph, that is, figure 1a in paragraph 3.1.1, figure 1b in paragraph 3.1.2 and figure 1c in paragraph 3.1.3.
Figures have been modified as recommended.
Line 280: The supplementary figure related to FTIR spectra of pure MCA and Cs powders is the S1 and not the S2.
The figure number has been corrected.
Line 287: Please add the characteristics of zeta potential analysis.
More information has been added to the experimental section.
Line 302: How the Authors justify the sentence “this did not modify their permeability to water”? Do they have some data about water permeability of blank gauzes?
We did not measure water permeability, we measured the air permeability
Lines 314-322: I think it is necessary adding some references for these statements.
References have been added.
Paragraph 3.2 This part is a little bit confuse. Firstly, the name of each samples (C-H2.5-Ag, C-H5-Ag, etc.) should be reported in paragraph “Method” the first time they are described in order to allow a better understanding of the text. The whole structure of the paragraph should be changed describing the samples in the same order they were prepared.
The labels have been highlighted in the experimental section. To study the effect of HAp content, HAp NPs of different concentrations (2.5 wt.% and 5 wt.%) were deposited on unmodified (H2.5 and H5), cationized (C-H2.5 and C-H5), and anionized (A-H2.5 and A-H5) cotton gauze samples.
Line 370: The analyzed sample (blank, Ag, H2.5-Ag, H5-370 Ag, C-H5-Ag, M-H5-Ag) should be also reported in the relative paragraph of “Materials and Methods” section. Moreover, the sample called “Ag” was not described in that section, thus it is not clear how it was obtained and its nature.
The labels have been highlighted in the experimental section. To study the effect of HAp/Ag content, HAp/Ag NPs of different HAps concentrations (2.5 wt.% and 5 wt.%) were deposited on unmodified (H2.5-Ag and H5-Ag), cationized (C-H2.5-Ag and C-H5-Ag), and anionized (A-H2.5-Ag and A-H5-Ag) cotton gauze samples.
Lines 374-376: The SEM images demonstrate that the HAp/Ag NP are formed; however, the difference of HAp/Ag NP amounts onto the surfaces between the different samples, as the Authors affirm, is not clear. Could the Authors add other images, even in different magnification, in order to clarify it?
We are sorry due to COVID pandemic we can not provide more images.
Lines 385-387: This sentence does not coincide with the images: “Elemental mapping by EDS showed areas of higher HAp NP (phosphorus and calcium) content in the cationized samples (Cs) than in the anionized samples and in the blank”. Elemental mapping reported by the Authors showed a difference in calcium content between cationized and blank sample, but there is no obvious difference with anionized one.
The sentence was corrected. The data extracted from the TGA and SEM-EDS analyses (Figure S3) showed areas of higher HAp NP (phosphorus and calcium) content in the cationized samples (Cs) than in the anionized samples and in the unmodified cotton gauze (blank).
Table 1: According to the Authors statement the anionized sample has higher AgNps than cationized one but, in the table, the percentage of Ag in the A-H5-Ag sample is less than C-H5-Ag. These data are in contradiction. I strongly suggest to Authors to check the data and correct the table or the text.
The TGA and SEM-EDX analyses (Table 1) confirmed that compared with the unmodified cotton gauze (blank), the cationic (Cs) modification increased HAp NP deposition on the cotton gauze, while the anionic modification (carboxylation or deposition of HAps NPs) mainly increases Ag NP deposition.
Line 428: The figure 5 is already reported in the text, please correct it in figure 6.
Figure number and location has been corrected.
Line 442: The figure 5c should be moved in the paragraph 3.5.
Figure number and location has been corrected.
Line 450: The figure 5d should be moved in the paragraph 3.5.
Figure number and location has been corrected.
Table 2: The table should be moved below the paragraph 3.6.
The table has been moved.
Reviewer 2 Report
In this article by Said and co-workers, the authors studied the formation of multifunctional hydroxyapatite/silver nanoparticles/cotton gauze fabrics and investigated their antimicrobial performance. The work itself is quite extensive and definitely fits the scope of Nanomaterials. However, the article should be subjected to extensive editing to make it suitable for publication in this journal. Please find the suggestions below:
1) The reference formatting does not follow the MDPI requirements. Please correct it.
2) Please sharpen the description of the novelty factor in the "in this work" section of the introduction. What exactly was done in this study for the first time? It is important to put this article in perspective to enable readers to quickly decide whether to read the paper or not.
3) Headlines should not be separated from the corresponding sections (Line 146)
4) Sonication is a strong form of agitation, which causes cavitation. That is why it is always important to specify the parameters of the process such as power and amplitude. Please supplement it in the revised version of the manuscript.
5) Other experimental parameters are missing such as (the list is non-exhaustive): SEM acceleration voltage, the gas flow rate in TGA, etc. Please carefully screen the Experimental section and include all the necessary details to reproduce this study. Without enabling others to validate the study, this will have a reduced impact as researchers will not be able to build on these findings.
6) Formatting of the manuscript should be considerably improved. Examples of corrections to be made:
- there is lots of empty space on Pages 6, 8, 11-15, 17-19
- the plots are not of the same size. Some of them additionally are too large
- on the other hand, Fig. 6 contains a substantial amount of text which is too small to read
7) Overall, the article appears too long as much of the data could be moved to the SI file. Please consider migration of the non-essential information to the supplement to increase the clarity of this contribution, which will make it more approachable.
8) Why the Ag micrograph in Fig. 3 is not at the same magnification as all the other images?
Author Response
1) The reference formatting does not follow the MDPI requirements. Please correct it.
Authors: The reference formate has been corrected to fit the MDPI.
2) Please sharpen the description of the novelty factor in the "in this work" section of the introduction. What exactly was done in this study for the first time? It is important to put this article in perspective to enable readers to quickly decide whether to read the paper or not.
Authors: The introduction (the last paragraph) has been revised and more information was added.
3) Headlines should not be separated from the corresponding sections (Line 146)
The Headlines have been reordered
4) Sonication is a strong form of agitation, which causes cavitation. That is why it is always important to specify the parameters of the process such as power and amplitude. Please supplement it in the revised version of the manuscript.
Authors: The sentences had been revised “Then, cotton gauze samples (10×10 cm2) were put in the prepared Cs solution (liquor ratio = 1:30) and sonicated at power 40W, 50 °C for 30 min, followed by drying at 90 °C for 5 min, and curing at 120 °C for 1 min.”
Authors: More information has been added.
5) Other experimental parameters are missing such as (the list is non-exhaustive): SEM acceleration voltage, the gas flow rate in TGA, etc. Please carefully screen the Experimental section and include all the necessary details to reproduce this study. Without enabling others to validate the study, this will have a reduced impact as researchers will not be able to build on these findings.
Authors: The gas flow rate and SEM accelerating voltage have been added to the characterization section.
6) Formatting of the manuscript should be considerably improved. Examples of corrections to be made:
- there is lots of space on Pages 6, 8, 11-15, 17-19
- the plots are not of the same size. Some of them additionally are too large
- on the other hand, Fig. 6 contains a substantial amount of text which is too small to read
Authors: The paper/figure formate has been improved.
7) Overall, the article appears too long as much of the data could be moved to the SI file. Please consider migration of the non-essential information to the supplement to increase the clarity of this contribution, which will make it more approachable.
Authors: non-essential information has been added to the supplementary information.
8) Why the Ag micrograph in Fig. 3 is not at the same magnification as all the other images?
Authors: Figures formate has been improved.
9) The title of the manuscript should be more relevant, representative.
Authors: The title has been reworded
10) Keywords. The keywords should be more precise, taking into account that they should represent the content of the manuscript and be specific to field or sub-field. Abstract. The Abstract section should be revised by inserting some key results while keeping only the relevant information/data.
The keywords have been reworded
The abstract has been improved.
Reviewer 3 Report
The title of the manuscript should be more relevant, representative.
Keywords. The keywords should be more precise, taking into account that they should represent the content of manuscript and be specific to field or sub-field.
Abstract. The Abstract section should be revised by inserting some key results, while keeping only the relevant information/data.
Generally, the Introduction section should clarify the motivation for the work presented and prepares readers for the structure of the paper. The Introduction should give some background information and set the context. In this regard, the authors should rewrite/ shorten the introduction section (literature review) in order to present the (recent) state of the art in the field of manuscript. Furthermore, the novelty of the paper should be underlined.
Results and Discussion. The authors do not provide any comparison with previously works of similar materials (in all cases/ characterization techniques). It will be better if a relative comparison can be made.
Figure 2. Please rescale the y axis of Figure 2c to 10-20o.
Figure 4. Please use the chemical symbol instead of chemical name for carbon, silver, etc.
Figure 3c. Please replace “roching” with “rocking”.
Conclusions. Please reformulate this section by highlighting the novelty.
References. There is a tendency of self-citation. This should be reconsidered. Also, the number of references should be reduced.
All the text must be revised to improve the English language.
I consider that the article can be accepted for publication only after a major revision.
Author Response
The title of the manuscript should be more relevant, representative.
The title has been reworded
Keywords. The keywords should be more precise, taking into account that they should represent the content of the manuscript and be specific to field or sub-field.
The keywords have been reworded
Abstract. The Abstract section should be revised by inserting some key results while keeping only the relevant information/data.
The abstract has been improved.
Generally, the Introduction section should clarify the motivation for the work presented and prepares readers for the structure of the paper. The Introduction should give some background information and set the context. In this regard, the authors should rewrite/ shorten the introduction section (literature review) in order to present the (recent) state of the art in the field of the manuscript. Furthermore, the novelty of the paper should be underlined. Results and Discussion. The authors do not provide any comparison with previous works of similar materials (in all cases/ characterization techniques). It will be better if a relative comparison can be made.
Some recent studies have been added to the introduction section
Figure 2. Please rescale the y axis of Figure 2c to 10-20o.
The Figure has been modified as recommended.
Figure 4. Please use the chemical symbol instead of the chemical name for carbon, silver, etc.
The Figure has been modified as recommended.
Figure 3c. Please replace “roching” with “rocking”.
The Figure has been modified as recommended.
Conclusions. Please reformulate this section by highlighting the novelty.
The conclusion has been reworded.
References. There is a tendency for self-citation. This should be reconsidered. Also, the number of references should be reduced.
References have been revised and reduced.
All the text must be revised to improve the English language.
The English language has been checked and corrected.
I consider that the article can be accepted for publication only after a major revision.
Round 2
Reviewer 1 Report
I appreciate the changes made by the Authors, since now the manuscript seems to be clear and complete. However, I still would like to add some other comments, for minor issues.
Line 168-169: still, it is not clear how Hap NPs were obtained at 2.5 wt.% and 5 wt.% concentrations.
Line 312: Since the Authors do not measure water permeability, how they could be sure that shrinkage and to smaller holes in the cotton fabrics does not modify it?
Line 471 and 536: both the paragraphs are counted as 7. Please check and correct the numeration.
As solution, Authors could unify them in just one paragraph.
Author Response
Authors: we appreciate your kind support.
Respond to: Line 168-169: still, it is not clear how Hap NPs were obtained at 2.5 wt.% and 5 wt.% concentrations.
The paragraph has been modified as fellow: “To study the effect of HAp content, the HAp NPs were deposited on the surface of cotton gauze at different concentrations (2.5 wt.% and 5 wt.% based on the dry weight of the unmodified cotton gauze). The HAp cotton gauze samples are as fellow: unmodified (H2.5 and H5), cationized (C-H2.5 and C-H5), and anionized (A-H2.5 and A-H5) cotton gauze samples.
Respond to: Line 312: Since the Authors do not measure water permeability, how they could be sure that shrinkage and to smaller holes in the cotton fabrics does not modify it?
The paragraph has been modified as fellow: The tensile strength was higher in the HAp-Ag-Cotton gauze (540±3 N/cm2) than in the blank cotton gauze (510±3 N/cm2), due to the effect of the Cs and carboxymethyl modifications and HAp and Ag NP coating. These results showing that ultrasonic irradiations and experimental conditions did not damage cotton gauze structure to a significant level. His results agreed with Noman and Petrů [60] who modified the cotton samples with ZnO NPs using an in situ ultrasonic acoustic method and alkaline treatment. Aside from tensile properties, we have noticed slight shrinkage in the cotton gauze samples upon washing. Generally speaking, the cotton gauze fibers contain large amounts of -OH groups and they are highly hydrophilic [61]. Cellulose fibers in the cotton gauze exhibit a low degree of crystallization, so that when cotton gauze absorbs water, the bonding force among cellulose fibers (chains) is reduced markedly, which causes swelling. Therefore, when cotton fabrics are twisted and undergo plastic deformation. Consequently, the fabric shrinks and wrinkles [61].
Respond to: Line 471 and 536: both the paragraphs are counted as 7. Please check and correct the numeration.
The paragraph has been modified as fellow: Elemental composition and HAp and Ag NPs distribution on the surface of the different samples were studied by SEM-EDX with elemental mapping (Figure 7). The Ag-coated cotton gauze was made mainly of C, O, and Ag, and HAP/Ag-coated cotton gauze was composed of C, O, Ca, P, and Ag. The data extracted from SEM-EDS analyses (Figure S3, supplementary information) showed areas of higher HAp NP (Ca and P) in the cationized samples (Cs) than in the anionized samples and in the unmodified cotton gauze (blank).
Respond to: As a solution, the Authors could unify them in just one paragraph.
We unified them in just one paragraph.
Reviewer 2 Report
I agree with Reviewer #3 that it is unacceptable to pretend that corrections were made when they were not. The following comments of mine have been disregarded: (2) and (8). Please refer to them.
Author Response
I agree with Reviewer #3 that it is unacceptable to pretend that corrections were made when they were not. The following comments of mine have been disregarded: (2) and (8). Please refer to them.
Respond to 2: Please sharpen the description of the novelty factor in the "in this work" section of the introduction. What exactly was done in this study for the first time? It is important to put this article in perspective to enable readers to quickly decide whether to read the paper or not.
The paragraph has been reworded: Nowadays, antimicrobial and UV protective medical textiles are mainly obtained by coating the cellulose fabric with functional NPs. Considering the importance of surface modification and insitu synthesis of composite NPs onto medical cotton gauze, the present study aimed to develop colored, antimicrobial, and UV-blocker cotton gauze fabrics by coating them with HAp NPs by wet precipitation of HAp NPs followed by Ag NP phytosynthesis using ginger oil as a reducing agent [38]. Chitosan was used in surface modification due to it is biodegradability, biocompatibility, and most importantly, promotes wound healing, features that make it suitable as a starting material for wound dressings. The cotton gauze samples were modified by the addition of Cs (cationic modification) or by partial carboxymethylation (anionic modification) to further improve HAp and Ag NPs dispersion and adhesion to the fiber surface. The HAp/Ag NPs coating provides good UV-protection and antimicrobial properties. The Ag NPs gave very high antimicrobial activities against different microorganisms (i.e. Candida albicans, E. coli, and S. aureus), and a brilliant yellow-brown coloration compared with the unmodified cotton gauze sample. The innovative strategy involved three distinct steps: (1) Cationization of cotton gauze by reacting it with chitosan or anionization of cotton gauze through partial carboxymethylation using monochloroacetic acid. (2) Thus anionic and cationic modified cotton gauze along with unmodified samples (blank) was submitted to in situ formations of Ag NPs using Ginger oil which has been used as a reducing agent and stabilizing agent to prevent aggregation of Ag NPs and linker for fixation of Ag NPs on the surfaces of the cotton gauze. Furthermore, Ginger oil has been chosen as it is a green reducing agent for Ag NPs synthesis and for treating inflammatory conditions and their associated pain. It is recognized as safe to use by the US Food and Drug Administration (FDA) with systemic and local (skin) anti-inflammatory effects [39].
Respond to 8: Why the Ag micrograph in Fig. 3 is not at the same magnification as all the other images?
Sorry for the misunderstanding, the Ag NPs are smaller than HAps NPs, and to clearly observe and measure the particle size distribution of the Ag NPs. We apologies we do not have TEM images at larger scales.
Reviewer 3 Report
I consider that the article can be accepted for publication only after a major revision considering that the author did not responded to all previous recommendations.
1. Results and Discussion. The authors do not provide any comparison with previously works of similar materials (in all cases/ characterization techniques). It will be better if a relative comparison can be made.
2. References. There is a tendency of self-citation (example Barhoum - 7 citations). This should be reconsidered. Also, the number of references should be reduced.
Author Response
Respond to Results and Discussion. The authors do not provide any comparison with previous works of similar materials (in all cases/ characterization techniques). It will be better if a relative comparison can be made.
More information and discussion have been added: Sections 3.7 to 3.9
Respond to There is a tendency of self-citation (example Barhoum - 7 citations). This should be reconsidered. Also, the number of references should be reduced.
The number of refs has been reduced. We believe our references (Barhoum et al. citations) contain information valuable for the readers. Most of these papers are often cited.
Round 3
Reviewer 2 Report
I accept the paper for publication.
Author Response
Thank you for accepting our paper, English has been improved.
Reviewer 3 Report
The manuscript can be accepted after correct/ re-numbering subchapters and the format according to journal requirements.
Author Response
we have used the journal template and fit with journal requirements.